# Synthesis and Antimicrobial Properties of a Ciprofloxacin and PAMAM-dendrimer Conjugate

**DOI:** 10.3390/molecules25061389

**Published:** 2020-03-18

**Authors:** Søren Wedel Svenningsen, Rikki Franklin Frederiksen, Claire Counil, Mario Ficker, Jørgen J. Leisner, Jørn Bolstad Christensen

**Affiliations:** 1Department of Chemistry, University of Copenhagen, Thorvaldsensvej 40, DK-1871 Frederiksberg C, Denmark; swsvenningsen@gmail.com (S.W.S.); claire.counil@orange.fr (C.C.); mario.ficker@gmail.com (M.F.); 2Department of Veterinary and Animal Sciences, Food Safety and Zoonoses, University of Copenhagen, Grønnegårdsvej 15, DK-1870 Frederiksberg C, Denmark; rikki.fredriksen@umu.se (R.F.F.); jjl@sund.ku.dk (J.J.L.)

**Keywords:** antimicrobial compounds, dendrimers, Ciprofloxacin

## Abstract

Infections caused by bacteria resistant to antibiotics are an increasing problem. Multivalent antibiotics could be a solution. In the present study, a covalent conjugate between Ciprofloxacin and a G0-PAMAM dendrimer has been synthesized and tested against clinically relevant Gram-positive and Gram-negative bacteria. The conjugate has antimicrobial activity and there is a positive dendritic effect compared to Ciprofloxacin itself.

## 1. Introduction

Antimicrobial resistance to antibiotics is an increasing problem worldwide. Accordingly, the World Health Organization (WHO) has listed antibiotic resistance as one of the three most important public health challenges in the 21st century [1]. Two major causes for the present situation are the over prescription of antibiotics leading to resistance and the lack of new antibiotics. Bacterial resistance occurs via different mechanisms including degradation of the antimicrobial compound, modification of the antimicrobial target, decrease in drug uptake, global regulatory changes in metabolic pathways, and the activation of efflux mechanisms [2,3]. Resistance due to the expression of efflux-pumps is problematic, because such pumps are not very specific with respect to antibiotics, but are capable of exporting a wide variety of different classes of antimicrobial compounds, potentially causing multi-resistance [4,5,6,7,8,9].

A number of compounds which are inhibitors of efflux-pumps are known, and many of the classical drugs ranging from antihistamines to CNS-active compounds from psychiatry have this interesting side-effect, i.e., they are efflux-pump inhibitors at low concentrations but kill bacteria at higher concentrations [10,11,12,13,14]. Apart from the effect on the central nervous system, a serious issue is that the known efflux-pump inhibitors lack specificity for bacterial efflux pumps, but are also inhibitors for pumps found in humans, like glycoprotein P, and can interfere with the hERG-channel in the heart lengthening the QT-interval, increasing the risk of cardiac arrest [15,16].

A possible solution to efflux-mediated resistance could be modifications of known antibiotics so they are not substrates for efflux-pumps while keeping the antimicrobial activity.

D’Emanuele and coworkers [17] discovered that the beta-blocker propranolol covalently conjugated to a PAMAM-dendrimer was internalized by human cells. This was interesting, because propranolol itself is known to be a substrate for the glycoprotein P pump.

Since all families of efflux pumps [8] are highly promiscuous with respect to substrates, this led us to the idea of using a conjugate between a dendrimer and an antibiotic to enter the bacteria and prevent export by their efflux pumps.

The conjugation of an antibiotic to a dendrimer can be either non-covalent or covalent: Non-covalent attachment requires the formation of a guest–host complex between the antibiotic and the dendrimer that is sufficiently stable to survive the gastrointestinal system and in the blood stream, while either having an antimicrobial effect as a complex or releasing the antibiotic once inside the bacteria.

Xu and coworkers [18] studied the solubilization of the fluoroquinolones Nadifloxacin and Prulifloxazin in amino-terminated ethylenediamine-core PAMAM-dendrimers (EDA-core PAMAM) and found no effect on the solubility at low pH, which shows that a non-covalent complex could dissociate in the stomach and undergo uptake as two separate entities. In vitro, there was an enhanced antibacterial effect of combining Prulifloxacin with the G4-dendrimer (64 NH2-groups at the surface) against *E. coli*. Wen and coworkers [19] found that the complex between amino-terminated EDA-core PAMAM-dendrimers and Sulfomethoxazole showed enhanced activity in vitro against *E. coli*. There have also been in vitro studies on the effect of co-administering antibiotics with poly(propylene imine) dendrimers showing similar results [20]. Antimicrobial dendrimers are also known and some examples are the quaternary silicon-dendrimers pioneered by de la Mata, Gomez and coworkers [21,22,23,24] the small peptide-dendrons by Urbanczyk-Lipkowska and coworkers [25,26,27,28] and the peptide dendrimers by Reymond and coworkers [29,30,31,32].

Covalent conjugates between antibiotics have been described by Kannan and coworkers [33], who synthesized a covalent conjugate between Azithromycin and a hydroxyl-terminated generation 4 PAMAM-dendrimer that released the antibiotic after cellular uptake enabling treatment of intracellular *C. trachomatis* in vitro. Baker and coworkers [34] described a photocleavable conjugate between a dendrimer and Ciprofloxacin that released the drug upon illumination.

In the present work, we wanted to investigate if a dendrimer-antibiotic conjugate could have antimicrobial activity in itself and not just be an advanced and expensive drug-delivery system. It was important to choose an antibiotic, where the mechanism of activity was non-covalent interactions with a target, because in that way it might be possible to kill many bacteria subsequently with the same molecule in a “catalytic fashion”. The conjugate should optimally be active both towards Gram-positive and Gram-negative bacteria. The dendrimer would have to be small, because access to the interior of Gram-negative bacteria takes place through porines in the outer membrane that in general excludes larger molecules from passage [35,36].

The fluoroquinolones (Figure 1) are often used for treatment of Gram-negative infections and a lot is known regarding the effect of substitution on the ring systems. The carboxylic acid in position 3 and the carbonyl group in position 4 are essential for activity. Substitution in position 2 also leads to reduced activity, while the substituent in position 1 controls specificity against Gram negative or Gram positive bacteria. Position 7 is also important for specificity as well as serum half-life [37].

## 2. Results

Ciprofloxacin (1) (Figure 2) was chosen because it is active both against Gram-positive and Gram-negative bacteria and because it acts through non-covalent binding to topoisomerases. Ciprofloxacin targets primarily topoisomerase II in Gram-negatives and primarily topoisomerase IV in Gram-positives. It has furthermore a natural handle for attachment of a linker (the piperazine moiety) that should have minimal effect on the antimicrobial properties. A G0 DAB-core PAMAM-dendrimer [38] (Figure 2) was chosen as a compromise between multivalency, solubility of the conjugate and a wish to keep the molecular weight as low as possible.

The molecular design is shown in Figure 3 and involves the covalent attachment of Ciprofloxacin via a linker to a small PAMAM-dendrimer.

The synthesis is shown in Scheme 1 and started with protection of the piperazine in Ciprofloxacin with a BOC-group converting the zwitterionic compound into the carboxylic acid (**6**) followed by conversion into the fully protected ester (**7**), which was much easier to handle. Following deprotection of the BOC-group, compound (**8**) was reacted with Phenyl acrylate [39] to give (**9**). Phenyl esters have very high selectivity for acylation of primary versus secondary amines [40,41,42], which is why Michael-addition is observed instead of acylation. Reaction of phenyl ester (**9**) with the dendrimer (**2**) gave the benzyl ester (**10**), which was deprotected by catalytic hydrogenation (H_2_/Pd(OH)_2_) to give the desired product (**4**). The control compound (**3**) was synthesized by a similar route (Scheme 2), but using benzyl acrylate [43] for the Michael addition and deprotection as for (**4**).

The dendrimer-control was synthesized by acetylation of the dendrimer (**2**) with excess acetic anhydride.

The compounds were tested against four different strains of clinically relevant pathogens for which antibacterial resistance is a challenge representing two Gram-positive (*Enterococcus faecalis* and *Staphylococcus aureus*) and two Gram-negative species (*Pseudomonas aeruginosa* and *Salmonella enterica*). The *S. aureus* 8325-4 strain is a plasmid cured variant of the NTCT 8325 variant, an important model strain used in laboratory research [44,45]. The *E. faecalis* V583 strain (ATCC 700802, GenBank AE016830.1) is a vancomycin-resistant, clinical isolate from human blood [46,47]. The *S. enterica* serovar Typhimurium 4/74 strain represent a serovar commonly implicated in human gastroenteritis that is also used as a mouse model of human typhoid fever [48]. The *P. aeruginosa* PAO1 strain is a commonly used research strain for this opportunistic pathogen [49,50]. MIC-values for these target strains were determined by microbroth dilution assay in 96-well microtiter plates. The results are shown in Table 1 and the minimal inhibitory concentrations (MIC values) are given both in mg/L and in µmol/L.

Similar trends were observed for all four target organisms. However, our variant of the 4/74 strain exhibited an unusual lack of sensitivity towards Ciprofloxacin with only 0.8% of the *Salmonella* strains in the EUCAST database on antimicrobial wild type distributions exhibiting a similar or a higher MIC value (https://mic.eucast.org/Eucast2/). The strain SL1344 that is a derivative of the 4/74 strain [48] has been reported as Ciprofloxacin sensitive [51,52]. Although the 4/74 variant used in our study can in fact be described as resistant towards Ciprofloxacin it showed the same overall pattern in sensitivities as the three other target species that were all sensitive to Ciprofloxacin according to the EUCAST database.

The control dendrimer (**5**) lacked antimicrobial activity whereas the dendrimer Ciprofloxacin conjugate (**4**) showed highest activity in terms of MIC values expressed in µM showing a clear positive dendritic effect, which becomes even bigger if the MIC values are calculated relative to the number of Ciprofloxacines. This result emphasizes the strong synergistic activity of the dendrimer Ciprofloxacin conjugate. In conclusion, this compound contributes to the overall antimicrobial activity of the conjugate that is of interest as it moves its application beyond that of being a drug delivery vehicle.

## 3. Discussion

Successful penetration of the bacterial cell envelope is a prerequisite for the activity of antibiotics having intracellular targets. At physiological pH, fluoroquinolones are known to coexist in a zwitterionic and overall uncharged form and a neutral form, with only the latter form passively diffusing through cell membranes, such as the cytoplasmic membrane of Gram-negatives and Gram-positives [53]. For Gram-negatives, however, the outer membrane and its lipopolysaccharide (LPS) layer constitute a second barrier. Penetration by fluoroquinolones of this barrier is suggested to be mediated by three mechanisms: (i) by a hydrophilic pathway through porins, (ii) by a hydrophobic pathway through the lipid bilayer, and (iii) by a “self-promoted” pathway involving displacement of divalent cations bridging adjacent LPS molecules [54] such as seen for antimicrobial peptides. In hydrophilic fluoroquinolones, such as Ciprofloxacin, uptake is mediated primarily by porins and by the “self-promotion” pathway [55,56]. With Ciprofloxacin having a mass of 331 Da it is easily imagined that this hydrophilic molecule can cross the water-filled porins as they generally have an exclusion size of 600 Da [57]. However, the larger size of our dendrimer conjugate (876 Da), and the absence of classical porins in *Pseudomonas aeruginosa* [58], suggests that these compounds are instead transported by either the “self-promotion” pathway or by hijacking of specific channels and high affinity receptors translocating larger nutrient molecules [57].

The mode of action of Ciprofloxacin is by stabilizing the complex of prokaryotic topoisomerase II and IV enzymes, leading to DNA fragmentation and bacterial death [59,60]. The control dendrimer showed in our study no antimicrobial activity which is in contrast with studies on PAMAM dendrimers with terminated amino groups [61,62,63]. This result is most likely due to our control dendrimer was equipped with terminal acetyl groups. The lack of antimicrobial activity of the control dendrimer compared to the synergistic activity of the dendrimer Ciprofloxacin conjugate relative to Ciprofloxacin suggest that the synergism is related to Ciprofloxacin mode of action. Here, it is of interest that PAMAM dendrimers promote transfection of DNA in both mammalian cells [64] and in prokaryotic cells as demonstrated for *Anaplasma phagocytophilum*, an obligate intracellular bacterium [65]. Thus, the ability of PAMAM to bind DNA indicates that the dendrimer may interact with the association of Ciprofloxacin with bacterial topoisomerases e.g., by binding to DNA fragments generated by Ciprofloxacin activity.

We have not in our study compared the synergy effect observed for the dendrimer Ciprofloxacin covalent conjugate with the antimicrobial activities of mixtures of the dendrimer and Ciprofloxacin. A synergy effect has also been observed for mixtures of PAMAM and quinolones or sulfamethoxazole but this result appeared to be a concentration-dependent effect of the PAMAM dendrimer on the solubility of the antimicrobial compounds. For the mixture of PAMAM and quinolones this effect was also pH dependent. It was hypothesized that this could be due to electrostatically interactions between PAMAM and the carboxyl group in the quinolone molecules and in addition that both quinolones and sulfamethoxazole might be kept in the cavities of the PAMAM dendrimer resulting in increasing solubility of the small guest molecules [18,19]. Thus, the synergy mechanism in mixtures appears to differ from the mechanism for the covalent conjugate hypothesized above. This would make it difficult to interpret the outcome of a potential comparative analysis.

It will be of interest to examine further the mode of action of the dendrimer Ciprofloxacin conjugate and to illuminate whether this compound may show antimicrobial activity towards Ciprofloxacin resistant bacterial variants. Thus, it should be tested whether resistance due to mutations in the protein targets of Ciprofloxacin might be circumvented by the potential DNA binding abilities of the dendrimer. It is more straightforward to anticipate the overcome of resistance due to efflux pumps by using PAMAM as the vehicle to transport Ciprofloxacin to the topoisomerase targets but this also need to be tested experimentally. Such studies are in progress in our laboratories.

In conclusion, we have demonstrated proof of principle that the dendrimer Ciprofloxacin conjugate exhibits increased antimicrobial activity and that PAMAM dendrimers have potential in antimicrobial chemotherapy beyond acting as delivery vehicles. Such synergistic effects have been described before for anionic/cationic dendrimers and levofloxacin, another fluoroquinolone in a non-covalent system [66].

## 4. Experimental

### 4.1. Bacterial Cultures

*Salmonella enterica* serovar Typhimurium 4/74, *Pseudomonas aeruginosa* PAO1, *Staphylococcus aureus* 8325-4 and *Enterococcus faecalis* V583 from the local collection at the Institute were grown overnight with shaking (180 rpm) in 5 mL Luria-Bertani (LB) broth at 37 °C for 18 h before use for MIC determinations.

### 4.2. MIC Determinations

MIC was determined by microbroth dilution assay in 96-well microtiter plates with LB broth containing two-fold serial dilutions of test compounds. Test compounds were diluted in sterile water and prepared for a testing concentration range of 0.25–256 μg/mL. Then, 50 μL from each dilution was transferred into the well of a microtiter plate and inoculated with 50 μL of bacterial culture prepared for a final concentration of approximately 106 CFU/mL corresponding roughly to OD600 0.00065. The microtiter plates were then incubated aerobically at 37 °C for 18–24 h with shaking (180 rpm). Growth in individual plate wells was defined as OD600 > 0.06. Wells containing only inoculums were used as positive controls for bacterial growth, while wells with only media was used as negative control for sterility of media.

The MIC values are the median of three biological replicates, and demonstrated good reproducibility.

### 4.3. Synthetic Procedures

#### 4.3.1. 7-(4-(2-Carboxyethyl)piperazin-1-yl)-1-cyclopropyl-6-fluoro-4-oxo-1,4-dihydroquinoline-3-carboxylic acid (**3**)

To a round bottom flask equipped with a magnetic stirring bar was added compound **8** (1.6 g, 3.79 mmol) and dissolved in dichloromethane (40 mL). Benzyl acrylate (960 mg, 4.19 mmol) was added and the reaction was stirred at RT for 72 h under nitrogen atmosphere. The yellow solution was evaporated in vacuo, and dried under high vacuum. The pale-yellow powder was washed with Et_2_O (30 mL) and further dried in high vacuum. The intermediate compound (**11**) was obtained as pale white solid (1.05 g), which was used directly for the next step.

Compound (**11**) (1.0 g; 17.1 mmol) was dissolved in a 1:1 mixture of ethanol and THF (50 mL in total), 10% Pd(OH) _2_ on carbon (150 mg) was added and the mixture hydrogenated at 3 bar pressure at room temperature for 16 h. The reaction mixture was filtered through Celite to remove the catalyst, and compound (**3**) was obtained by removing all volatiles in vacuum. Yield: 580 mg (37%).

^1^H NMR (300 MHz, DMSO-d6): δ = 1.20–1.28 (m, 2H), 1.34–1.43 (m, 2H), 2.81–2.92 (m, 2H), 2.38–2.48 (m, 8 H), 2.57–2.58 (m, 2H), 3.85–3.95 (m, 1H), 7.60–6.69 (m, 1H), 7.92–8.05 (m, 1H), 8.68–8.75 (m, 1H). ^13^C NMR (126 MHz, DMSO-d6): δ 176.30; 172.87; 165.86; 153,91; 151.93; 147.87; 144.14; 139.09; 118.69; 111.03; 106.69; 62.57; 52.58; 52.00; 51.64; 48.55; 47.11; 46.26; 42.40; 36.48; 35.93; 30.88; 7.54. ^19^F NMR (470 MHz, DMSO-d6): δ −121.57. HRMS: ESI (0.1% Formic acid in MeOH): 404.161 (calcd. for C_20_H_22_FN_3_O_5_^+^: *m*/*z*: 404.162).

#### 4.3.2. DAB-PAMAM-G0-(Cipro-COOH)_4_ ● 6HCl (**4**)

Compound (**10**) (0.27 g; 0.11 mmol) was dissolved in a mixture of THF and water (10:3 *v*/*v*; 60 mL) and added to 10% Pd(OH)_2_ on carbon (0.10 g) in a hydrogenation bottle. The mixture was hydrogenated in a Parr-apparatus at 3.5 bar for 48 h. The catalyst was removed by filtration through a bed of Celite and concentrated in vacuo. A solution of HCl in Methanol (1 M, 30 mL) was added and the solution evaporated again. The hydrochloride was purified by size-exclusion chromatography on Sephadex G25 (Pharmacia (Sweden) now GE Healthcare (US) using Methanol:H_2_O (1:1 *v*/*v*) as eluent. After removal of the solvent, the product was obtained in a yield of 0.12 g (46%).

^1^H-NMR (500 MHz, DMSO-d6): δ 15.10 (s, 3 H); 8.69–8.61 (m, 4 H); 7.96–7.84 (m, 4 H); 7.65–7.53 (m, 4 H); 3.89–3.82 (m, 4 H); 3.41–3.23 (m, 48 H); 3.14 (s, 16 H); 3.11–3.06 (m, 8 H); 2.90–2.85 (m, 3 H); 2.71–2.67 (m, 8 H); 1.80–1.73 (m, 4 H); 1.37–1.29 (m, 8 H); 1.22–1.13 (m, 8 H). ^13^C-NMR (126 MHz, DMSO-d6): δ 176,29; 169.17; 165.79; 153.80; 151.80; 148.08; 139.03; 119.11; 111.18; 111.00; 106.80; 106.71; 51.56; 50.88; 48.61; 39.52; 38.41; 38.28; 36.49; 35.98; 29.39; 25.13; 7.63. ^19^F-NMR (470 MHz, DMSO-d6): δ −121.67. HRMS: ESI (0.1% Formic acid in MeOH): *m*/*z* [M + 2H^+^]^2^^+^: 1044.0084 (calcd. for C_104_H_134_F_4_N_22_O_20_^2+^: 1044.0052); *m*/*z* [M + 3H^+^]^3+^: 696.3405. (calcd. for C_104_H_135_F_4_N_22_O_20_^3+^: 696.3392); *m*/*z* [M + 4H^+^]^4+^: 522.0033. (calcd. for C_104_H_136_F_4_N_22_O_20_^3+^: 522.5062).

#### 4.3.3. DAB-PAMAM-G0-(Acetamide)_4_ ● 2HCl (**5**)

DAB-PAMAM-G0-(NH_2_)_4_ [34] (1.00 g; 1.84 mmol) was dissolved in ethanol (25 mL). Acetic anhydride (7.50 g; 73.5 mmol) was added and the reaction mixture was stirred at room temperature for 16 h. The reaction mixture was evaporated in vacuo, dissolved in 1 M HCl in Methanol (25 mL) and evaporated again. The residue was redissolved in 1 M HCl in Methanol (10 mL) and added dropwise to Diethyl ether (200 mL) under stirring. The precipitated product was filtered and dried in vacuum to give a white crispy solid. Yield: 1.44 g (quantitative). Ninhydrin-test for residual primary amino groups (1% Ninhydrin in ethanol): Negative after 24 h.

^1^H-NMR (500 MHz, D_2_O): δ 3.45-3.39 (m, 8 H); 3.25 (s, 20 H); 2.78–2.69 (m, 8 H); 1.93 (s, 12 H). ^13^C-NMR (126 MHz, D_2_O): δ 174.37; 171.86; 52.46; 49.60; 38.68; 28.74; 28.70; 21.82; 20.54. HRMS (MALDI, matrix: Dithranol): *m*/*z* [M+H^+^]^+^: 713.4682 (calcd. for C_32_H_60_N_10_O_6_: 713.4686); [M + Na^+^]^+^: 735.4503 (calcd. for. C_32_H_60_N_10_NaO_6_^+^: 735.4503); [M + K^+^]^+^: 751.4155 (calcd. for. C_32_H_60_N_10_KO_6_^+^: 751.4277).

#### 4.3.4. Benzyl 7-(4-tert-butoxycarbonyl)piperazin-1-yl)-1-cyclopropyl-6-fluoro-4-oxo-1,4-dihydroquinoline-3-carboxylate (**7**)

To a flame dried round bottomed flask equipped with a magnetic stirring bar was added: Ciprofloxacin (**1**) (1.50 g; 4.53 mmol); 1,4-dioxane (25 mL); Boc_2_O (1.19 g; 5.45 mmol) and Et_3_N (0.76 mL; 5.45 mmol). The reaction mixture was stirred at room temperature for 3 h and evaporated to dryness *in vacuo*. The residue was dissolved in acetonitrile (50 mL) and benzyl chloride (0.69 g; 5.45 mmol), K_2_CO_3_ (0.75 g; 5.43 mmol) was added. This reaction mixture was refluxed for 10 h and evaporated *in vacuo* to give a yellow powder, that was washed with heptane (100 mL), water (100 mL) and heptane (100 mL) followed by drying in high vacuum. White powder. Yield 2.23 (94%).

^1^H-NMR (500 MHz, CDCl_3_): δ 8.50 (s, 1 H); 8.00 (d, J = 13.2 Hz, 1 H); 7.50 (d, J = 7.5 Hz, 2 H); 7.36 (t, J = 7.4 Hz, 2 H); 7.30 (t, J = 7.3 Hz, 2 H); 5.37 (s, 2 H); 3.65 (t, J = 5.1 Hz, 4 H); 3.47–3.41 (m, 1 H); 3.22 (t, J = 4.8 Hz, 4 H); 1.49 (s, 9 H); 1.28 (q, J = 6.9 Hz, 2 H), 1.10 (q, J = 6.6 Hz, 2 H). ^13^C-NMR (126 MHz, CDCl_3_): δ 172.94; 165.63; 154.72; 154.48; 152.50; 148.44; 144.51; 138.15; 136.48; 128.64; 128.11; 128.05; 123.18; 113.55; 113.37; 110.07; 105.32; 80.34; 66.56; 50.10; 34.81; 28.54; 8.29. ^19^F-NMR (470 MHz, CDCl_3_): δ −123.48. HRMS: ESI (0.1% Trifluoroacetic acid in MeOH): *m*/*z* [M + H^+^]^+^: 522.2356 (calcd. for C_29_H_33_FN_3_O_5_^+^: 522.2399); *m*/*z* [M + 2H^+^]^2+^: 261.1219 (calcd. for C_29_H_33_FN_3_O_52_^+^: 522.2399).

#### 4.3.5. Benzyl 1-cyclopropyl-6-fluoro-4-oxo-piperazin-1-yl)-1,4-dihydroquinoline-3-carboxylate (**8**)

Compound (**7**) (0.53 g; 1.02 mmol) was dissolved in a mixture of trifluoroacetic acid and dichloromethane (2:8 *v*/*v*, 5 mL) and stirred at room temperature for 24 h. The reaction mixture was diluted with dichloromethane and washed with saturated aqueous NaHCO_3_ to remove the acid. The organic phase was dried over MgSO_4_, filtered and evaporated in vacuo to give the deprotected amine (**8**) as a powder. Yield: 0.39 g (90%).

^1^H-NMR (500 MHz, CDCl_3_): δ 8.47 (s, 1 H); 7.95 (d, J = 13.2 Hz, 1 H); 7.49 (d, J = 7.0 Hz, 2 H); 7.36 (d, J = 7.6 Hz, 2 H); 7.31–7.27 (m, 1 H), 7.23 (d, J = 7.1 Hz, 1 H); 5.35 (s, 2 H); 3.42–3.37 (m, 1 H); 3.29–3.22 (m, 4 H); 3.13–3.07 (m, 4 H); 2.59 (s, 1 H); 1.29–1.23 (m, 2 H), 1.11–1.06 (m, 2 H). ^13^C-NMR (126 MHz, CDCl_3_): δ 173.20; 165.46; 154.49; 152.51; 148.33; 138.07; 136.54; 128.60; 128.07; 128.01; 113.33; 113.14; 109.94; 104.93; 77.16; 66.40; 50.88; 45.88; 34.68; 8.22. ^19^F-NMR (470 MHz, CDCl3): δ −123.50. HRMS: ESI (0.1% Trifluoroacetic acid in MeOH): *m*/*z* [M + H]^+^: 422.1875 (calcd. for C_24_H_25_FN_3_O_5_^+^: 422.1875).

#### 4.3.6. Benzyl 1-cyclopropyl-6-fluoro-4-oxo-7-(4-(3-oxo-3-phenoxypropyl)piperazin-1-yl)-1,4-dihydroquinoline-3-carboxylate (**9**)

To a flame dried round bottomed flask equipped with a magnetic stirring bar was added: Compound (**8**) (1.00 g; 2.37 mmol), dichloromethane (25 mL) and phenyl acrylate [35] (0.3885 g; 2.62 mmol). The mixture was stirred under N_2_ for 72 h and evaporated in vacuo. The residual powder was washed with diethyl ether (20 mL) and dried in vacuum giving compound (**9**) as a white powder. Yield: 1.27 g (94%).

^1^H-NMR (500 MHz, CDCl_3_): δ 8.47 (s, 1 H); 7.95 (d, J = 13.3 Hz, 1 H); 7.50 (d, J = 7.0 Hz, 2 H); 7.39–7.34 (m, 4 H); 7.31–7.27 (m, 1 H); 7.24–7.20 (m, 2 H); 7.11–7.08 (m, 2 H); 5.36 (s, 2 H); 3.41–3.37 (m, 1 H); 3.30 (t, J = 4.8 Hz, 4 H); 2.93 (t, J = 6.9 Hz, 2 H); 2.81 (t, J = 7.1 Hz, 2H); 2.79–2.75 (m, 4 H), 1.27–1.22 (m, 2 H); 1.10 (m, 2 H). ^13^C-NMR (126 MHz, CDCl_3_): δ 173.08; 170.90; 165.47; 154.40; 152.42; 150.76, 148.34; 138.03; 136.56; 129.53; 128.59; 128.04; 127.97; 125.97; 122.99; 122.93; 121.59; 113.28; 113.10; 109.99; 104.95; 77.16; 66.38; 53.50; 52.74; 49.90; 34.65; 32.60; 8.18. ^19^F-NMR (470 MHz, CDCl_3_): δ −123.41. HRMS: ESI (0.1% Trifluoroacetic acid in MeOH): *m*/*z* [M+H^+^]^+^: 570.2415 (calcd. for C_23_H_33_FN_3_O_5_^+^: 570.2399); *m*/*z* [M + 2H^+^]^2+^: 285.6224 (calcd. for C_29_H_33_FN_3_O_5_^2+^: 285.6236); *m*/*z* [M + Na^+^]^+^: 592.2235 (calcd. for C_23_H_33_FN_3_NaO_5_^+^: 592.2219); *m*/*z* [M + K^+^]^+^: 608.1961 (calcd. for C_23_H_33_FN_3_KO_5_^+^: 608.7204).

#### 4.3.7. DAB-PAMAM-G0-(Cipro-Bn)_4_ (**10**)

In a flame dried round bottomed flask equipped with a magnetic stirring bar was compound (**8**) (0.50 g; 0.88 mmol) dissolved in dry DMSO (10 mL). The solution was heated to +40 °C and 4-(Dimethylamino)pyridine (0.0108 g; 0.088 mmol) was added followed by dropwise addition of a solution of DAB-PAMAM-G0-(NH_2_)_4_ [34] (0.100 g; 0.184 mmol). The reaction mixture was stirred for 4 days at 40 °C under a N_2_-atmosphere and then transferred to a dialysis bag (Regenerated cellulose, molecular weight cut-off 1 kD) and dialyzed against 0.2 M HCl (2 × 2 L, 24 h each) followed by milliQ-water (2 × 2 L, 24 h each). The content of the dialysis bag was then transferred to a round bottomed flask and lyophilized to give the benzylester protected dendrimer conjugate (**10**), which contains encapsulated phenol as evidenced from NMR. Yield 0.36 g (80%). Ninhydrin-test for residual primary amino groups (1% Ninhydrin in ethanol): Negative after 24 h.

^1^H-NMR (500 MHz, DMSO-d6): δ 8.44 (d, J = 19.7 Hz, 4 H); 8.38–8.20 (m, 5 H, included phenol); 7.73 (m, 4 H); 7.51–7.45 (m, 8 H); 7.45–7.42 (m, 3 H); 7.42–7.36 (m, 12 H); 7.34–7.30 (m, 4 H); 5.28–5.24 (m, 8 H); 3.67–3.62 (m, 4 H); 3.52–3.28 (m, 24 H); 3.25–3.17 (m, 8 H); 3.13 (s, 16 H); 3.10–2.85 (m, 24 H); 2.85–2.73 (m, 4 H); 2.66–2.58 (m, 8 H); 1.75–1.65 (m, 3 H); 1.26–1.21 (m, 8 H); 1.11–1.06 (m, 8 H). ^13^C-NMR (126 MHz, DMSO-d6): δ 171,55; 171.47; 164.48; 164.44; 153.53; 153.38; 151.57; 151.42; 150.46 (phenol); 148.42; 137.99; 137.92; 136,61; 129.49 (phenol); 128.35; 127.73; 127.58; 125.81 (phenol); 122.11; 121.76 (phenol); 111.70; 111.52; 108.79; 106.35; 65.22; 52.34; 51.90; 51.74; 51.54; 51.39; 49.33; 48.72; 48.27; 39.52; 38.45; 38.24; 36.47; 34.84; 30.41; 7.53. ^19^F-NMR (470 MHz, DMSO-d6): δ −124.29. HRMS: ESI (0.1% Formic acid in MeOH): *m*/*z* [M + 2H^+^]^2^^+^: 1224.1091 (calcd. for C_132_H_158_F_4_N_22_O_20_^2+^: 1224.0991); *m*/*z* [M + 3H^+^]^3+^: 816.4073. (calcd. for C_132_H_159_F_4_N_22_O_20_^3+^: 816.4018; *m*/*z* [M + 4H^+^]^4+^: 612.5532). (calcd. for C_132_H_160_F_4_N_22_O_20_^4+^: 612.5532).

NMR & MS spectra of the compounds are available in Appendix A.

## 5. Conclusions

Ciprofloxacin has been conjugated to a DAB-core G0 PAMAM-dendrimer and the antimicrobial activity has been tested on a selection of both Gram-positive and Gram-negative bacteria of clinical relevance. A positive dendritic effect is observed for the conjugate in both types of bacteria. This is an example of a dendrimer that adds to the antimicrobial activity of a conjugate which could lead to new strategies for the treatment of infectious diseases.

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
