# Peer review of "Synthesis and Antimicrobial Properties of a Ciprofloxacin and PAMAM-dendrimer Conjugate"

_molecules, 2020, doi:10.3390/molecules25061389_

Round 1

Reviewer 1 Report

the comments are in the attached file

Author Response

RE: It is necessary that the authors give the data of the elemental analyses of each compound in order to determine the purity of them.

any proton or carbon NMR spectrum of the derivatives are shown in the manuscript. At least the spectra of the conjugates PAMAM-ciprofloxacin should be added to the manuscript.

Answer: We can only agree, but a major problem with polar dendrimers like PAMAM-dendrimers is, that complete removal of residual solvents if often close to impossible due to the formation of guest-host complexes. We have added all the spectroscopic information at hand as supplementary material. The NMR-work has been done on an instrument equipped with a cryoprobe, which means that even tiny amounts of impurities are detectable. I can add, that we have recently been able to get crystalline DAB-core PAMAM G0 (the starting material) but it still contains crystal solvent (water or methanol), so it is not a request, that is easily solved.

RE: In figure 2 and 3, the structure of the PAMAM dendrimer must be drawn correctly with all its branches in the correct position. (Compound 2 and 5)

Answer: This has been corrected.

RE: The authors conjugate the antibiotic to the dendritic skeleton by covalent bond. How much MIC would be obtained if instead of using the covalent bond they used the physical mixture of both compounds? Perhaps treatment of the bacteria with a physical mixture of ciprofloxacin and dendrimer would also lead to a synergistic effect

Answer: We did not look a non-covalent conjugates because guest-host complexes between dendrimers and polar compounds are usually not very stable but dissociates as soon as pH gets below 7 meaning that the dendrimer and the ciprofloxacin would be taken up individually. The acetylated dendrimer itself did not show any antimicrobial activity, but there could be an effect on bacterial uptake of other compounds, which will be interesting to investigate. 

RE: Page 6 line 2, the word ciprofloxacin is duplicate 

Answer: This has been corrected.

Reviewer 2 Report

The authors present a very good work on the synthesis of a dendrimer conjugate and antimicrobial activity.

The only think that I would rise a question is about the coincident results for MIC mg/L and μmgM  (16 - 0.0069) for three cell lines comprising Gram positive and negative bacteria. This should be better explained.

I also would advise to complement the conclusions with a few more words on perspective for new conjugate dendrimers to act in same fashion of the studied ones. This can enrich the work with more in future approaches to achieve antimicrobial molecules.

Author Response

Regarding the comments from Reviewer 2:

RE: The only think that I would rise a question is about the coincident results for MIC mg/L and μmgM  (16 - 0.0069) for three cell lines comprising Gram positive and negative bacteria. This should be better explained.

This is a good point. We have discussed it, and is it unlikely in view of having measured 16 MIC values from 12 different concentrations, that 3 of them are similar?

RE: I also would advise to complement the conclusions with a few more words on perspective for new conjugate dendrimers to act in same fashion of the studied ones. This can enrich the work with more in future approaches to achieve antimicrobial molecules.

We have added more text to the discussion of the results. With respect to future approaches, I think work needs to be done to understand uptake of dendrimers in bacteria.   

Round 2

Reviewer 1 Report

The changes introduced in the manuscript are correct and appropriate. I agree with the authors that obtaining elementary analyses of dendritic systems is complicated in most cases and that it retains solvent. Nevertheless, and taking into account the spectra supplied in the supplementary material, I consider the characterization of the systems to be correct.
Therefore, I recommend the article for its publication with the new changes introduced.